# Fungal Stress Responses and the Importance of GPCRs

**DOI:** 10.3390/jof11030213

**Published:** 2025-03-11

**Authors:** Daniela Lara-Martínez, Fabiola Estefania Tristán-Flores, Juan Antonio Cervantes-Montelongo, Guillermo Antonio Silva-Martínez

**Affiliations:** 1Posgrado de Ingeniería Bioquímica, Departamento de Ingeniería Bioquímica y Ambiental, Tecnológico Nacional de México en Celaya, Celaya 38010, Guanajuato, Mexico; d2203015@itcelaya.edu.mx (D.L.-M.); fabiola.tristan@itcelaya.edu.mx (F.E.T.-F.); 2Departamento de Ciencias Básicas, Tecnológico Nacional de México en Celaya, Celaya 38010, Guanajuato, Mexico; 3Departamento de Ingeniería Bioquímica y Ambiental, Tecnológico Nacional de México en Celaya, Celaya 38010, Guanajuato, Mexico; 4Escuela de Medicina, Universidad de Celaya, Celaya 38080, Guanajuato, Mexico; 5Investigadores por México (IxM) CONAHCYT—Tecnológico Nacional de México en Celaya, Celaya 38010, Guanajuato, Mexico

**Keywords:** G-protein-coupled receptors (GPCRs), fungal stress response, gene regulation, environmental adaptation, signal transduction

## Abstract

G-protein-coupled receptors (GPCRs) play a crucial role in the gene regulation of processes related to the response to different types of stress in fungi. These receptors act as sensors of extracellular signals and transmit the information to the interior of the cell through G-proteins. In the presence of different and specific types of stresses, GPCRs activate signaling cascades that culminate in the activation of transcription factors, which regulate the expression of genes associated with the stress response, including those induced by changes in environmental pH. GPCR-mediated gene regulation allows fungi to adapt to adverse conditions such as osmotic, thermal, oxidative, or nutritional stress, as well as fluctuations in environmental pH. This review focuses on the understanding of how GPCRs modulate the stress response in fungi and their crucial role in advancing our knowledge of the physiology and adaptability of these microorganisms in their changing environment.

## 1. Introduction

All living organisms can adapt to various environmental changes. Sensing these environmental signals and ensuring an appropriate cellular response is critical for survival. Every cellular response begins with an extracellular stimulus that is sensed by a plasma membrane component and then translated into a signal that can be interpreted within the cell, resulting in an adaptive physiological response [1,2]. Stress can include almost any environmental condition that differs from the optimal growth condition of the organism under study [3]. Stress adaptation depends on three fundamental principles: (i) the ability to sense environmental changes; (ii) the ability to transduce these signals to regulate cellular processes; and (iii) the adaptive responses themselves, which allow cells to survive stress [4].

In fungi, adaptation to a changing environment, which includes hosts, natural habitats, and culture media, is essential for their success. Fungi possess complex signaling pathways that allow them to respond appropriately to fluctuations in external stimuli, such as biotic or abiotic stressors. These stressors include changes in temperature, low nutrient levels, pH, oxygen limitation, ultraviolet radiation, and oxidative and osmotic stress. These fluctuations can disrupt cellular homeostasis and cause molecular damage [5]. Therefore, fungi must be able to adapt to these dynamic changes to survive, grow, and colonize any niche. Adaptation in fungi involves adjusting to stress or modifying gene expression patterns in response to environmental signals. G-protein-coupled receptors (GPCRs) play a crucial role in the gene regulation of processes related to the response to different types of stress in fungi. These receptors act as sensors of extracellular signals and transmit the information to the interior of the cell through G-proteins. In the presence of different and specific types of stresses, GPCRs activate signaling cascades that culminate in the activation of transcription factors that regulate the expression of genes associated with the stress response, including those induced by changes in environmental pH. GPCR-mediated gene regulation allows fungi to adapt to adverse conditions such as osmotic, thermal, oxidative, or nutritional stress, as well as fluctuations in environmental pH. This review explores the intricate interplay between GPCRs and G-proteins and how this dynamic interaction sets in motion a wide range of intracellular signaling pathways. Understanding this relationship is particularly crucial given the significance of fungi and their pivotal role in various biological processes.

## 2. G-Protein Coupled Receptors (GPCRs)

Receptors are integral membrane proteins located on the cell surface, acting as molecular recognition sites. They facilitate the binding of various molecules, triggering specific intracellular responses. These receptors are primarily categorized into three types, each operating through a unique mechanism: enzyme-linked receptors, initially recognized for their role in responding to extracellular signal proteins that promote growth, proliferation, differentiation, or survival of cells in animal tissues [6]; ion channel receptors, which regulate the cell’s homeostatic activity and its programmed responses to both intra- and extracellular stimuli [7]; and G-protein-coupled receptors.

G-protein coupled receptors (GPCRs) are a type of protein in the cell membrane known for their ability to detect external signals and regulate a series of processes within the cell when these signals activate them. This happens because when an external molecule binds to a GPCR, it causes changes in the protein’s shape, which in turn activates a chain of events within the cell to produce a specific response [8,9,10].

All GPCRs share a fundamental structural blueprint composed of seven transmembrane domains (TMs), each comprising 19 to 30 hydrophobic amino acids. These domains cross the plasma membrane and are connected by three extracellular loops containing highly conserved cysteine residues that form disulfide bonds to stabilize the receptor structure, and three intracellular loops, as shown in Figure 1 [1,2,11].

GPCRs in fungi are exceptionally diverse in terms of the signals they can perceive. They can detect signals such as pheromones, hormones, proteins, peptides, nutrients, ions, environmental stress, and light. This diversity allows fungi to coordinate their activities, metabolism, and development. Ultimately, this helps fungi survive, reproduce, and be virulent in certain contexts. GPCRs in fungi are crucial for these organisms to respond effectively to their environment [12,13,14,15,16,17].

## 3. GPCR Activation and Signaling

G-protein-coupled Receptors (GPCRs) act as intermediaries between changes and cellular responses at the transcriptional level. They consist of an extracellular N-terminal region and an intracellular C-terminal portion that harbors phosphorylation sequences and palmitoylation sites through reactive cysteines. These intracellular sites are vital for interaction with heterotrimeric G-proteins, triggering complex signaling cascades [18,19].

GPCRs play a crucial role in transmitting signals from the external environment into the cell through a well-coordinated process (Figure 2). Initially, these receptors are situated in the cell membrane, featuring extracellular domains capable of interacting with specific signaling molecules, known as ligands. These ligands, which may include neurotransmitters, hormones, or environmental chemicals, induce a change in the structural balance of GPCRs towards active configurations upon binding. This promotes the subsequent interaction with signal transducers, such as heterotrimeric G-proteins (Gαβγ), at the receptor’s inner core and sets off diverse cellular responses [1,20].

Upon exposure to external stimuli, GPCRs initiate a sequence of conformational changes that activate G-proteins. The alterations in the GPCR structure diminished affinity for guanine nucleotide binding to the Ras-like α subunit within the heterotrimeric G-protein complex. These rearrangements lead to three crucial events in GPCR functioning. First, a reduction in the affinity of the Gα subunit for GDP occurs, resulting in the dissociation of GDP. Subsequently, the binding of GTP stabilizes a conformational change in the Gα subunit, activating it. Secondly, there is a detachment of the βγ subcomplex from the G-protein. Lastly, the G-protein disengages from the GPCR [19,21]. 

### 3.1. Gα and Gβγ Subunit Functions

The Gα subunit plays a crucial role in regulating intracellular signaling pathways in fungi, serving as a vital modulator or transducer in diverse transmembrane signaling systems. It is engaged in a broad spectrum of biological processes, including vegetative growth, the development of infection-related structures, asexual conidiation, and virulence [22].

The activation of the Gα subunit, by exchanging GDP for GTP, results in its detachment from the βγ subunits and subsequent initiation of various intracellular signaling pathways [23]. This process is tightly regulated by GTPase-activating proteins (GAPs) and guanine nucleotide exchange factors (GEFs) governing the GDP/GTP exchange on the Gα subunit [24]. 

In the context of fungal development and pathogenicity, the Gα subunit impacts a diverse array of phenotypic outcomes, as shown in Table 1. For example, in *Aspergillus flavus*, Regulator of G-protein signaling (RGS proteins), functioning as negative regulators of Gα subunits, play significant roles in governing conidia, sclerotia, and aflatoxin formation, indicating the involvement of the Gα subunit in fungal development and secondary metabolite production [25]. 

Different subtypes of the Gα subunit, such as Gαs, Gαi, Gα12/13, and Gαq, play a decisive role in determining the cellular response to a specific signal. These subtypes regulate effectors such as adenylate cyclase, Rho/Guanine nucleotide exchange factors (GEF), and phospholipase Cβ (PLCβ), thereby influencing the course of intracellular signaling. On the other hand, the Gβγ subunit initiates its own independent signaling pathway [26,27].

The Gβγ subunit plays a versatile role in modulating intracellular signaling pathways in fungi, affecting various physiological and developmental processes. Within the realm of filamentous fungi, Gβγ has been implicated in regulating Golgi fragmentation during mitosis, a process vital for proper cell division and development. Reduction or inhibition of Gβγ results in delayed mitotic progression and diminished Golgi fragmentation upon microtubule disruption, indicating its involvement in maintaining cellular structure and responding to stress [28]. Furthermore, in *Botrytis cinerea*, knockout mutants of the G-protein β subunit gene Bcgb1 displayed altered development and virulence, along with changes in cAMP signaling and MAPK pathway activation, suggesting that Gβγ influences both cAMP and MAPK signaling pathways [23]. Additionally, Gβγ subunits have been identified as regulators of anterograde and retrograde trafficking of proteins, modulating the function of intracellular organelles and transcriptional events, thus reshaping our understanding of GPCR function and Gβγ signaling in fungi [29]. Emerging evidence also suggests that Gβγ may play roles in intracellular compartments such as endosomes, the Golgi apparatus, and the nucleus, orchestrating a broad spectrum of cellular activities [30]. Moreover, the intrinsic disorder of Gγ subunits, a conserved structural feature among eukaryotes, is inherently critical for their signaling roles, with alterations in the Gγ tail structure affecting the stability of the Gγ subunit and its interaction with effectors, thereby impacting G-protein signaling [31]. 

When it is necessary to terminate a cellular response, the Gα subunit carries out the hydrolysis of GTP back to GDP, triggering the cessation of the response. Subsequently, the Gα subunit reassociates with the Gβγ dimer and the GPCR at the cell membrane, preparing to initiate the signaling cycle once again. This cyclic mechanism allows for precise control of cellular signaling events, ensuring meticulous regulation of biological responses [8,32].

**Table 1 jof-11-00213-t001:** Roles of Gα and Gβγ subunits in fungal signaling.

Organism	G-protein Subunit	Function	Ref.
*Aspergillus flavus*	Gα: Unknown	RGS proteins, functioning as negative regulators of Gα subunits, play significant roles in governing conidia, sclerotia, and aflatoxin formation, indicating involvement of Gα subunit in fungal development and secondary metabolite production.	[25]
*Metarhizium robertsii*	Gα: MrGPA2, MrGPA4	Gα subunits MrGPA2 and MrGPA4 contribute to vegetative growth, stress tolerance, and pest control potential, emphasizing their role in fungal adaptation and virulence.	[31]
*Botrytis cinerea*	Gα: Bcg1, Bcg2, Bcg3; Gβ: Bcgb1	Deletion of the Gβ subunit resulted in alterations in cAMP signaling and MAPK pathway activation, suggesting the critical role of Gα subunit-mediated signaling cascades in fungal development and virulence.	[33,34]
*Aspergillus fumigatus*	Gα: Unknown	Gα subunits regulate growth, germination, asexual development, and resistance to oxidative stress via PKA or PKC signaling pathways.	[31]
*Fusarium verticillioides*	Gβ: FvGbb2	Gβ-like protein, FvGbb2, regulates fumonisin biosynthesis, vegetative growth, conidiation, and stress response, highlighting diverse functions of Gβγ subunits in fungal physiology and pathogenicity.	[35]
*Magnaphorte oryzae*	Gα: magA, magB, magC; Gβ: mgb1; Gγ: MGG1	All three Gα subunit genes are involved in mating, but only magB, Gβ, and Gγ are required for appressoria formation and virulence.	[23]

### 3.2. Fungal GPCR Classification

The discovery of the different members of the GPCRs has been accompanied by the generation of a classification system based on sequence homology and structural similarity. GPCRs, common to all classification systems include pheromone receptors, receptors involved in nutrient, carbon, amino acid, and nitrogen sensing, receptors with similarity to cAMP receptors, and receptors with similarity to microbial opsins [1]. 

In addition to their primary role in signaling, heterotrimeric G-proteins exert fundamental control over a variety of biological processes, including growth, development, and virulence across a wide range of organisms, spanning from filamentous fungi to plants. In filamentous fungi, these proteins participate in critical processes such as cell growth, differentiation, sexual mating, sporulation, and pathogenicity. The contribution of individual subunits, such as Gα, Gβ, and Gγ, to these processes can vary depending on the species, and in some cases, their function is essential for virulence in plant pathogens [23,29].

In the field of research concerning GPCRs within fungi, an intricate categorization has been developed, encompassing a broad array of classes and subtypes. This categorization has evolved as our understanding of the functions and structural resemblances of these receptors across various fungal species has deepened. Displayed below is Table 2, which summarizes the primary groups of fungal GPCRs, accentuating recent updates that advance our comprehension in this domain. This table offers an encompassing view of the diverse GPCR landscape in fungi, serving as a foundational resource for exploring their roles in cellular signaling and various biological functions.

Recent studies have extended the classification of fungal GPCRs, incorporating new categories that enhance our knowledge of the diversity in fungal signaling. This expanded framework is based on the research led by Martín et al. [51]. The research team undertook a comprehensive genomic analysis of filamentous fungi and conducted functional and comparative assessments of GPCRs from a variety of organisms. Their efforts culminated in the development of a robust and intricate classification system that is grounded in the distinct nature and functional roles of the ligands interacting with GPCRs. Such advancements significantly deepen our understanding of fungal cellular signaling mechanisms and elucidate their roles in cell development and the biosynthesis of secondary metabolites [51].

In the new classification, the classical fungal GPCRs (corresponding to classes I–V and IX, including pheromone I, pheromone II, carbon, nitrogen, cAMP-receptor-like, and microbial opsin receptors) remain unchanged (Table 2). On the other hand, Table 3 presents the new categories (classes VI–VIII and X–XIV).

### 3.3. GPCRs and Signaling Pathways

In fungi, GPCR-regulated signaling pathways, including the cAMP-activated Protein Kinase A (PKA) pathway, the Mitogen-activated Protein Kinase (MAPK) cascade pathway, and the Phospholipase C (PLC) pathway, influence gene expression to regulate cell growth, morphogenesis, mating, stress responses, and metabolism in a complex and overlapping manner [15].

Fungi serve as valuable models for investigating eukaryotic signaling pathways, including those essential for adaptation to diverse environmental challenges encountered during host–pathogen interactions. One such pathway is the Pal/Rim alkaline response pathway, which enables fungi to sense and respond to fluctuations in ambient pH levels. Alongside other signaling cascades such as MAPK and cAMP/PKA, the Pal/Rim pathway is integral to the detection and adaptation to variations in nutrient availability, pH, oxygen tension, and host immune responses [53,54,55].

It is well known that GPCRs in fungi are vital for development and metabolic adaptations. Recently, it has been reported that in *Arthrobotrys flagrans*, which are nematode-trapping fungi, the GPCR GprC responds to nematode pheromones, regulating gene expression and boosting respiration through dual localization in the membrane and mitochondria; this dual role, such as in the human CB1 receptor, suggests an evolutionary conservation [56]. Also, *A. oligospora* uses GPCRs to detect nematode-derived pheromones (ascarosides), triggering trap formation via cAMP-PKA pathways. Two GPCR families—yeast glucose receptor homologs and Pth11-like receptors—mediate ascaroside-dependent and -independent sensing, respectively [57]. These findings highlight the multifunctionality and evolutionary significance of fungal GPCRs, underscoring their potential as therapeutic targets. 

## 4. Pal/Rim Pathway

While numerous signaling pathways undergo regulation by pH, one of the most specialized pathways is the Pal/Rim alkaline response pathway. Extensively examined across various fungal species, this pathway holds particular significance in fungal pathogenesis [55]. The Pal/Rim pathway facilitates the proteolytic activation of a transcription factor known as PacC in filamentous fungi or Rim101 in yeast. This factor modulates gene expression depending on the pH of the surrounding medium [58,59,60,61].

Essential for fungi’s adaptation to environmental pH, the Pal/Rim pathway encompasses a sophisticated signaling mechanism pivotal for growth, secondary metabolic processes, and pathogenic potential. Initially scrutinized in Ascomycota, it comprises seven members categorized into two complexes. Basidiomycota species primarily recognize homologs of the endosomal membrane complex and the transcription factor PacC/Rim101 [62]. PacC, a key player in this pathway, acts as a regulator for secondary metabolite production in fungi such as *Trichoderma harzianum*, where alterations in its expression influence the production of specific secondary metabolites and the fungus’s antifungal activity [63]. Similarly, in *Neurospora crassa*, the PAC-3 variant of PacC governs genes associated with cell wall synthesis, hydrolase activity, and fungal virulence, underscoring its significance in fungal adaptation to varying environmental conditions [64]. The regulatory mechanisms of PacC, extensively explored in *Aspergillus nidulans*, exhibit conservation across fungal species, including phytopathogenic fungi, where it governs developmental processes, pathogenicity, and mycotoxin biosynthesis [65].

Additionally, components of the Pal/Rim pathway, such as the novel Rho-like protein RHO4 in *Ustilago maydis*, contribute to its functionality by participating in proteolytic cleavage that activates PacC/Rim101, thereby influencing the fungus’s growth rate and stress responses [65]. Beyond fungal physiology, the Pal/Rim pathway, and its constituents, such as PacC, bear implications in bioremediation and stress resistance. For instance, *Dentipellis* sp. KUC8613 utilizes non-ligninolytic enzymes for Polycyclic Aromatic Hydrocarbon (PAH) degradation, a process potentially modulated by pathways involving PacC [66]. Moreover, genes such as ssk1 in *Trichoderma atroviride* suggest a regulatory role in response to various stresses, including osmotic and oxidative stress, which could be intertwined with the Pal/Rim pathway [67]. 

As shown in Figure 3, PalH/Rim21 is a membrane receptor with 7 transmembrane domains responsible for sensing the pH signal and transmitting it to the interior via its C-terminal tail, which interacts with PalF/Rim8, an a-arrestin. It is important to note that, although PalH/Rim21 has seven transmembrane domains, similar to GPCRs, it is not strictly a GPCR as its downstream partner is arrestin rather than a heterotrimeric G-protein. Moreover, CFEM domain-containing GPCRs, which are specific to fungi, are also noteworthy. A key question about these receptors is whether their signals are transduced by heterotrimeric G-proteins. CFEM domain-containing GPCRs do interact with heterotrimeric G-proteins, underscoring their importance in fungal signal transduction pathways. Given their fungal specificity, understanding how CFEM domain-containing GPCRs and G-proteins interact can provide deeper insights into fungal biology and signal transduction mechanisms [68,69]. Nonetheless, it plays a crucial role in the unique fungal PacC pathway. PalA/Rim20 then interacts with two YPXL motifs present in the PacC transcription factor; these motifs flank the sequence of the first proteolytic cleavage, which is pH-dependent and is performed by PalB/Rim13, a calpain-like protease. PacC then adopts an open conformation that allows access to the proteolytic process by the proteosome (in a pH-independent manner), which removes the C-terminal end of PacC, allowing PacC to act as an activator of gene expression at alkaline pH and as a repressor of genes expressed at acidic pH [64].

The Pal/Rim signaling pathway has been studied for a long time in fungi belonging to the division Ascomycota, until it was possible to identify, by homology, some of the components of this pathway in the Basidiomycota fungus *Ustilago maydis*; these include PacC, PalA, PalB, PalC, and PalI, but for Basidiomycetes, some of the components of the pathway that could be responsible for pH sensing at the membrane level are still unknown [64,65,70]. 

In the fungus *Ustilago maydis*, a gene encoding a small protein tentatively designated as RHO4 appears to be involved in the operation of the Pal/Rim pathway. When the RHO4 gene is deleted, the mutant displays a characteristic set of behavioral and phenotypic changes, such as those observed in mutants affected in the Pal/Rim pathway. For example, its growth capacity is strongly compromised and can even be completely inhibited in an alkaline environment. Additionally, it shows a high sensitivity to ionic and osmotic stress, and its ability to secrete proteases (enzymes that degrade proteins) is affected, while there are no changes in its ability to transition between yeast and mycelium forms [62].

The most compelling evidence of the involvement of the RHO4 gene in the Pal/Rim pathway lies in the fact that its mutation affects the second proteolytic event of the transcription factor PacC/Rim101. It is important to note that this second proteolytic event, which transforms an inactive intermediate form into the active form of the PacC/Rim101 transcription factor, is a rather poorly understood process. It is known that this occurs in both Ascomycota and Basidiomycota, two groups of fungi, but the protease enzyme responsible for it and its regulation are still unknown. The results obtained by Cervantes-Montelongo and Ruíz-Herrera [62], when studying the behavior of the Δrho4 mutant, could represent the first indication of a factor involved in the regulation of this important process.

In this pathway, GPCRs play a critical role as receptors of extracellular signals. When fungi are exposed to an alkaline environment, hydrogen ions (protons) are removed from the environment, resulting in an increase in extracellular pH. At this point, GPCRs come into play as they can sense this pH change and become activated in response to alkalinity.

## 5. MAPK Pathway

In fungi, the MAPK pathways are involved in different physiological and developmental processes, including cell cycle, mating, morphogenesis, sporulation, cell wall assembly and integrity, autophagy, pathogenesis, UV and heat-shock resistance, cell–cell signaling, fungus–fungus interactions, fungus–plant interactions, response to different forms of stress, response to damage-associated molecular patterns (DAMPs), etc. The MAPK signaling pathway in the yeast mating process is essential for transmitting the signal from the cell surface to the nucleus, where it regulates gene expression and coordinates cellular events necessary for cell fusion, followed by entry into meiosis for the formation of stress-resistant spores [54,71].

The activation of Mitogen-Activated Protein Kinase (MAPK) proteins in fungi (Figure 4) is a crucial process governing a diverse range of cellular reactions to internal and external stimuli, encompassing stress responses, development, and pathogenicity. MAPK signaling pathways are conserved across eukaryotes, playing essential roles in facilitating fungal adaptation to changes in the environment and interactions with hosts [34,72,73]. In the context of fungal pathogenicity, core regulators such as Pmk1, Hog1, and Slt2 have been identified among MAPKs. These proteins play roles in various aspects of fungal life, including hyphal development, asexual reproduction, and the ability to infect host plants [74,75]. For example, in *Cytospora chrysosperma*, MAPKs are indispensable for conidiation, stress responses, and virulence, with distinct yet intersecting functions observed across different MAPKs [76,77]. Similarly, in *Fusarium oxysporum*, ambient pH levels have been demonstrated to influence MAPK-mediated pathogenicity, illustrating the complex interplay between environmental signals and MAPK activation [73,78]. Moreover, MAPK signaling is implicated in regulating secondary metabolite production, which can influence fungal virulence and adaptation [79].

In the pheromone signaling of the yeast *S. cerevisiae*, it all starts when two types of pheromones, one soluble α-factor and another lipid-based a-factor, bind to their respective special GPCRs type receptors. These receptors are called Ste2 in MATa cells and Ste3 in MATα cells. When these pheromones bind to their receptors, they trigger a process that results in the release of a protein unit called Gβγ (composed of the proteins Ste4 and Ste18) from the receptor to which a Gα unit called Gpa1 is coupled. The released Gβγ anchors itself to the plasma membrane, which is the layer surrounding the cell. The liberated Gβγ plays a fundamental role in activating pheromone signaling. It directly activates the MAPK signaling cascade by bringing a MAPK signaling platform called Ste5 to the plasma membrane. Additionally, it binds to two proteins named Far1 and Cdc24. Far1 has an important role in regulating the progression of the cell cycle, while Cdc24 is a factor that activates another protein called Cdc42, which in turn plays an essential role in regulating cell growth and division [71].

The protein Ste5, located in the plasma membrane, is crucial for coordinating all this pheromone signaling. It acts as a central point by inducing the activation of a series of cascading proteins that form the MAPK signaling pathway. This cascade includes Ste11 (a Mitogen-Activated Protein Kinase Kinase Kinase, MAPKKK), Ste7 (MAPKK), and Fus3 (a Mitogen-Activated Protein Kinase, MAPK). Together, these proteins orchestrate a series of molecular events that ultimately lead to specific cellular responses to pheromones, triggering processes such as mate searching and reproduction in yeast [71].

In Table 4, we present some fungi species that have been studied along with their respective MAPKs and their main functions. These MAPKs play a crucial role in regulating cell integrity, responses to stress, reproduction, and the production of secondary metabolites in different fungal species.

## 6. cAMP/PKA Pathway

The cAMP signaling pathway in fungi is composed of several key components, including G-protein-coupled receptors (GPCRs), heterotrimeric G-proteins, adenylate cyclase (AC), cAMP-dependent protein kinase (PKA), and downstream effectors, such as transcription factors, as shown in Figure 4. This pathway serves as a crucial conduit for translating extracellular signals into intracellular responses. The process unfolds as follows: extracellular signals are detected by GPCRs, initiating a cascade of events. These events involve the activation of heterotrimeric G-proteins, which, in turn, stimulate adenylate cyclase (AC) to produce cAMP. Subsequently, cAMP activates PKA through a phosphorylation process. Once activated, PKA proceeds to phosphorylate downstream proteins, including transcription factors. This phosphorylation of various target proteins contributes to the regulation of a multitude of biological processes within the fungal cell [54,69].

The cAMP signaling pathway is a critical determinant of virulence in fungal plant pathogens, animal pathogens, and even malaria parasites. Additionally, it significantly influences the biocontrol abilities of organisms such as *Trichoderma* spp., *Metarhizium anisopliae*, and *Beauveria bassiana*. This pathway governs processes such as appressoria formation, the secretion of cell-wall-degrading enzymes, and the production of secondary metabolites [84].

*Trichoderma* has been extensively researched concerning the G-protein system’s functionality. Deletion experiments involving specific GPCR genes have revealed a substantial reduction in Trichoderma’s ability to combat and parasitize various pathogens. For example, silencing the GPCR-encoding gene gpr1 in *Trichoderma atroviride* resulted in a notable loss of its capacity to attack pathogens such as *Rhizoctonia solani*, *Botrytis cinerea*, and *Sclerotium sclerotium* [85].

Similarly, comprehensive insights into the roles of Gα-encoding genes within different Trichoderma species have been unveiled. The absence of the Gα-encoding gene tgaA in *Trichoderma virens* resulted in a significant reduction in its ability to antagonize *Sclerotium rolfsii*, accompanied by the loss of its capacity to parasitize the sclerotia of *S. rolfsii* [86].

Adenylate cyclases (ACs) play a central role within the cAMP signaling pathway, orchestrating the conversion of ATP into cAMP when activated by Gα. ACs exert regulatory control over various aspects of fungal life, spanning growth, development, mating, morphology, conidiation, metabolite production, and resilience to environmental stresses. Furthermore, the involvement of ACs in the biocontrol capabilities of microparasitic fungi, such as *Trichoderma* spp., has been prominently emphasized. For instance, deliberate removal of the adenylate cyclase-encoding gene tac1 in *T. virens* significantly impacted several important biological processes, including growth rate, morphological characteristics, sporulation, conidial germination, secondary metabolite production, and the capacity to effectively counteract pathogens such as *S. rolfsii*, *Rhizoctonia solani*, and *Pythium* sp. [87].

Protein kinase A (PKA), a serine/threonine kinase comprising regulatory and catalytic subunits, emerges as another essential component within this intricate signaling cascade. In the absence of cAMP binding, regulatory subunits interact with catalytic subunits, rendering them inactive. However, cAMP binding initiates a cascade of conformational changes within PKA. This leads to the binding of regulatory subunits to cAMP and the consequent release of catalytic subunits. Activated catalytic subunits then proceed to regulate gene expression by phosphorylating downstream targets, exerting a pivotal influence on an array of biological behaviors. PKA serves as a critical determinant in governing growth, development, metabolism, morphological characteristics, and the virulence and invasive potential of microorganisms [88].

Lastly, transcription factors (TFs) play a commanding role in numerous physiological processes within fungi. Their impact extends to crucial aspects such as growth, development, morphological characteristics, secondary metabolite production, and resilience to environmental stressors [88].

## 7. Adaptation Mechanisms of Fungi

Adaptation in fungi involves adjusting to stress or modifying gene expression patterns in response to environmental signals. This ability to adapt is crucial for their survival, growth, and colonization of various niches. Fungi are exposed to fluctuations in external stimuli, such as biotic or abiotic stressors, including changes in temperature, low nutrient levels, pH, oxygen limitation, ultraviolet radiation, and oxidative and osmotic stress. These fluctuations can disrupt cellular homeostasis and cause molecular damage. To respond appropriately to these dynamic changes, fungi possess complex signaling pathways that allow them to detect and react to these external stimuli [12,89,90].

G-protein-coupled receptors (GPCRs) are essential for fungi to adapt and survive in changing environments by allowing them to detect and respond to various environmental stimuli. These receptors act as crucial intermediaries in transmitting signals from the external environment into the cell, thereby coordinating cellular processes such as function, metabolism, and growth to enhance the survival, reproduction, and virulence of fungi [91]. GPCRs can detect a wide variety of molecules, including nutrients, pheromones, and environmental stress factors, enabling fungi to adapt to their ecological niches and overcome environmental challenges [92].

### 7.1. GPCRs and Their Association with Fungal Adaptation

Fungal stress resistance is evolving in a manner that is independent of fungal phylogeny; either way, fungal species that are closely related in phylogenetic terms do not necessarily display similar levels of stress resistance [93]. 

Environmental stress significantly impacts fungal resistance, affecting both their genomic plasticity and physiological responses. Fungi exhibit genomic changes such as polyploidy, aneuploidy, and copy number variation in response to stress, enhancing their fitness and resistance to antifungal drugs [94]. Extremophilic fungi, adapted to harsh environments, possess unique genes and pathways conferring resistance to abiotic stresses, making them valuable for enhancing stress tolerance in economically important plants and microbes [95]. Plants interact with beneficial fungi and bacteria to boost resistance against pathogens by modulating hormone synthesis and crosstalk, highlighting the interplay between biotic and abiotic stress responses in fungal communities [96].

*Candida auris*, an emerging pathogen, illustrates how environmental stress resistance contributes to its persistence and transmission in healthcare settings, despite sensitivity to current laundering protocols [97]. Fungi’s adaptation to heat stress is crucial for survival and pathogenicity, particularly with rising environmental and host temperatures during infection [98]. The HOG MAPK pathway regulates fungal adaptation to various environmental stresses, impacting growth, development, and pathogenicity [99]. Responses to chemical stress involve transcriptional changes and alterations in ergosterol biosynthesis genes, potentially leading to resistance mechanisms that spread in the environment [100].

Fungi’s resistance to chronic ionizing radiation is associated with resistance to chromium and elevated temperatures rather than acute radiation resistance, suggesting distinct stress resistance mechanisms [101]. Genetic modifications, such as inserting stress tolerance genes, induce species-specific physiological changes, emphasizing the importance of understanding fungal stress responses for industrial applications [102]. Finally, the wheat pathogen *Zymoseptoria tritici* demonstrates temperature-induced morphological transitions as part of a survival strategy in response to intracellular osmotic stress caused by heat stress, regulated by specific transcription factors and protein phosphatases [75].

#### 7.1.1. Osmotic Stress

When fungi encounter osmotic stress, such as high salt concentrations, GPCRs play a critical role by initiating signaling pathways that enable the accumulation of compatible solutes. This accumulation helps the fungi maintain osmotic balance and prevent excessive water loss. In the presence of oxidative stress, caused by the accumulation of reactive oxygen species, GPCRs activate signaling pathways that regulate the expression of genes involved in antioxidant defense. This protective mechanism protects the fungal cells from oxidative damage [4,89].

Managing changes in water balance is one of the fundamental challenges for fungi in most environments. Exposure to NaCl or KCl imposes osmotic stress, which causes rapid water loss, a reduction in cell size, and the loss of turgor presence. The fungus must restore its turgor pressure before it can resume growth, and to achieve this, it activates the synthesis and accumulation of intracellular osmolytes such as glycerol [4]. Table 5 showcases instances of fungi that have developed mechanisms to cope with osmotic stress.

The HOG signaling pathway plays a crucial role in regulating cellular responses to stress conditions related to salt concentration and extreme temperatures in the yeast *S. cerevisiae*. In the case of *Candida albicans*, Hog1 is tasked with regulating cell cycle progression in response to oxidative stress. Although historically primarily associated with the response to osmotic stress, it has been discovered that this pathway has multiple functions and is involved in a variety of important cellular processes [103,104].

**Table 5 jof-11-00213-t005:** Fungi with mechanisms for coping with osmotic stress.

Organism	Response to Osmotic Stress	Mechanisms and Pathways Involved	Key Findings	Ref.
*Setosphaeria turcica*	Alters mycelial growth and boosts conidia germination and yield.	Activation of HOG-MAPK pathway, up-regulation of aquaglyceroporin gene StFPS1.	Crucial for appressorium formation and penetration.	[105]
*Beauveria bassiana*	Influences carbohydrate utilization and stress sensitivity.	G-protein coupled receptor BbGPCR3, major MAP kinase pathways.	Integral to broad developmental and genetic networks.	[106]
*Saccharomyces cerevisiae*	Utilizes general stress response (GSR) and Unfolded Protein Response (UPR).	Involvement of Hac1p, Rpd3p, activation of GSR genes.	Protection against hyperosmotic stress.	[107]
*Saccharomyces cerevisiae*	Crosstalk between osmotic stress and GPCR signaling pathways.	Mediated by branched-chain amino acid metabolites and involvement of MAPKs and GPCRs.	Significance in stress adaptation.	[108]
*Saccharomyces cerevisiae*	Activates HOG pathway and glycerol synthesis for osmolarity adjustment.	Involvement of the Sho1 and Sln1 pathways.	Crucial for intracellular osmotic balance.	[109]
*Pichia pastoris*	Emphasizes stress-induced damage repair over glycerol accumulation.	Unique regulatory mechanism of the MAPK/HOG pathway.	Essential for survival under hyperosmotic stress.	[110]
*Aspergillus fumigatus*	Relies on glycerol biosynthesis, particularly via the G3PDH gene.	Crucial for adaptation to various stressors.	Importance in stress adaptation.	[89]
*Candida albicans*	Influences glycerol accumulation and osmotic stress response gene expression.	Mediated by SPT20 gene via Hog1-MAPK pathway.	Crucial for osmotic stress adaptation.	[78]
*Candida glabrata*	Importance of glycerophospholipid metabolism and membrane integrity	Interaction between Hog1 and transcription factor CgRds2.	Vital for cell survival under osmotic stress.	[103]
*Aspergillus flavus and Aspergillus ochraceus*	Regulates growth, development, and mycotoxin production under stress.	HOG-MAPK pathway, SLN1 and SHO1 branches, genes such as sln1, sho1, ste11, ssk2, pbs2, and hog1.	Controls aflatoxins and ochratoxins in agricultural products.	[111]
*Hortaea werneckii and Wallemia ichthyophaga*	Senses and responds to hypersaline conditions with unique adaptations.	HOG pathway, proteins involved in sensing high osmolarity, and MAP kinase module.	Structural differences in salt tolerance.	[112]
*Scedosporium apiospermum*	Activated by various stressors, including osmotic agents.	HOG pathway.	Acts as a general stress hub.	[113]
*Debaryomyces hansenii*	Essential for survival under high osmolarity, regulates glycerol and stress responses.	HOG1 pathway, mitogen-activated protein kinase Hog1.	Enhances halotolerance by regulating stress responses.	[114]

Yeast, such as *S. cerevisiae*, can continuously detect changes in its environment through osmolarity sensors located on the cell surface. These sensors allow yeast to monitor and adapt to changing environmental conditions, including changes in salt concentration. When environmental changes are detected, receptors on the cell surface transmit signals through GTPases in MAPK phosphorylation cascades. Once these cascades are activated, the process of gene transcription begins, leading to the production of proteins designed to cope with high osmolarity conditions [115].

In this context, Hog1 plays a central role in defining the cellular response. It acts as a key regulator in this signaling cascade, directing the activation of genes and proteins necessary for yeast to adapt and survive under osmotic stress conditions.

Furthermore, this signaling pathway, known as the HOG pathway, has been the subject of extensive research in various fungi and has been shown to play an essential role in adapting to high osmolarity conditions. In fact, the deletion of Hog-encoding genes in various fungi, such as *Bipolaris oryzae*, *Magnaphorte oryzae*, *Ustilaginoidea virens*, and *Fusarium graminearum*, results in an inability of these fungi to adapt to high salt concentration environments [105].

#### 7.1.2. Oxidative Stress

Oxidative stress in fungi is a critical factor that influences their survival, pathogenicity, and interaction with host organisms. It occurs when the cell’s production of reactive oxygen species (ROS) exceeds its ability to handle them with antioxidants, which normally keep the internal redox environment in balance. This redox balance is carefully maintained by a complex system of checks and balances, and small deviations from the normal level are tolerated only for a short time. If they persist longer, it is called “oxidative stress” or even “reductive stress”. Both conditions can lead to cell death by apoptosis or necrosis, which are programmed cell death processes [4,116].

Fungi must efficiently manage reactive oxygen species (ROS) to establish successful infections in host plants, as observed in *Alternaria alternata*, where treatment with H_2_O_2_ improves ROS metabolism and activates antioxidant defense mechanisms [117,118]. Similarly, Fusarium species show varied responses to H_2_O_2_-induced oxidative stress, affecting their growth and mycotoxin production, highlighting the role of H_2_O_2_ in modulating fungal secondary metabolism [119].

In the context of fungal pathogens such as *Aspergillus fumigatus*, an efficient ROS detoxification system is vital for survival within the host’s ROS-rich environment. The Oxr1 protein in *A. fumigatus* plays a crucial role in oxidative stress resistance by regulating catalase function, which is essential for the pathogen’s virulence and its interaction with the host’s immune response [115]. This underscores the potential of targeting oxidative stress pathways as a strategy to develop new antifungal treatments.

Furthermore, the genetic basis of oxidative stress tolerance in plant pathogens, such as *Zymoseptoria tritici*, involves complex networks and genetic architecture, where mapping of quantitative trait loci (QTL) has identified genomic regions associated with oxidative stress tolerance [116]. This suggests that inherent growth characteristics and specific unidentified genes within the major QTL contribute to oxidative stress tolerance in fungi. Cryptococcus neoformans mutants deficient in superoxide dismutase enzymes exhibit increased vacuolar fragmentation in response to oxidative stress, indicating the importance of vacuoles in fungal physiology and virulence under stress conditions [120].

Recent research has observed that in budding yeast, exposure to moderate stress conditions triggers a temporary adaptive response to higher levels of the same stress. This response leads to the acquisition of resistance to stressful situations, such as salt and oxidative stress, through the production of osmotic compounds and an enhancement in the elimination of reactive oxygen species (ROS). These response mechanisms could establish a state of resistance that helps prevent damage to the cells [121,122].

Additionally, in a study conducted by J. González et al. [122], it was found that the expression of a single gene (DhCTT1) in a strain of *S. cerevisiae* lacking the catalase enzyme allowed for rapid growth in the presence of ethanol (2%) and under conditions of salt stress (0.6 M NaCl), conferring high resistance to oxidative stress.

Regarding *Candida glabrata*, it has been discovered that this species rapidly detects and responds to changes in its metabolism and oxidative stress by activating protective enzymes against oxidative stress, such as catalase and superoxide dismutases (SODs), in addition to non-enzymatic defense systems such as glutathione (GSH). The regulation of the oxidative stress response involves transcription factors such as Msn2, Msn4, Skn7, and Yap1. The redundancy in these pathways suggests that the survival of *C. glabrata* depends on multiple routes that can compensate for each other [123].

#### 7.1.3. Heat Shock Stress

Heat shock stress induces a multifaceted response in fungi aimed at survival and adaptation. When confronted with elevated temperatures, fungi activate heat shock transcription factors and chaperones, coordinating cellular responses to mitigate heat-induced damage. This involves modifying cell membrane composition to maintain integrity and function, such as adjusting the balance between saturated and unsaturated fatty acids, and synthesizing heat shock proteins (HSPs) to safeguard against thermal damage [124]. Furthermore, heat stress significantly impacts fungal community dynamics, leading to shifts in competitive outcomes among species. This is partially attributed to changes in the secretion of inhibitory compounds and the synthesis of heat shock proteins, which may confer advantages to slower-growing species by allowing them more time to establish defensive mechanisms [125].

The pivotal role of small heat shock proteins (sHSPs) as cellular chaperones is evident in various cellular functions and stress responses, underscoring their potential as targets for antifungal therapy [126]. The intricate interplay between heat shock and cell wall integrity pathways is vital for fungal thermotolerance, with transcription factors such as HsfA playing indispensable roles in viability and adaptation to heat stress [127]. Despite the absence of traditional heat shock response elements in some fungi, robust upregulation of stress response genes indicates the presence of alternative adaptation mechanisms [128]. Moreover, heat stress triggers increased mobility of transposable elements in fungal genomes, contributing to genetic variation and potentially facilitating adaptation during infection [129]. In *Ganoderma lucidum*, heat stress induces elevated cytosolic Ca^2+^ levels, which regulate heat shock signal transduction and downstream responses, including HSP accumulation and secondary metabolite biosynthesis [130].

Heat shock transcription factors (TFs) play critical roles in the virulence of various fungi, such as *B. bassiana, M. robertsii*, and *Hirsutella minnesotensis*. Deletion or disruption of specific heat shock TF-encoding genes (Hsf1, Sfl1, Skn7 in *B. bassiana*; MrSkn7 in *M. robertsii*; SKN7 in *H. minnesotensis*) significantly impacts their ability to infect and harm their respective hosts. For example, in *B. bassiana*, the absence of these genes influences its virulence towards *G. mellonella* larvae. Similarly, in *M. robertsii*, the absence of the MrSkn7 gene affects its virulence towards wax moth larvae. This pattern is also observed in *Hirsutella minnesotensis*, where the disruption of the SKN7 gene weakens its ability to infect nematodes. In essence, these heat shock TFs are indispensable for the virulence mechanisms of these fungi against their respective hosts [131,132,133].

#### 7.1.4. pH

Microorganisms constantly face variations in environmental pH, prompting the development of regulatory systems to maintain homeostasis for survival, growth, differentiation, and secretion of various compounds essential for physiological functions [1,62].

Intracellular pH gradients play crucial roles in numerous cellular processes, with fungi exhibiting adaptive responses to environmental pH changes by regulating the secretion of enzymes, permeases, and secondary metabolites [1].

Fungi, such as *Aspergillus nidulans*, have evolved sophisticated mechanisms to thrive under extreme pH conditions, employing regulatory pathways mediated by zinc-finger transcription factors such as PacC, CrzA, and SltA to manage ambient alkalinity and saline stress [134,135]. These pathways modulate gene expression in response to external pH changes, triggering distinct transcriptional responses to different stress conditions [136,137]. In the context of human infection, *Aspergillus fumigatus* faces a combination of acidic/alkaline stress and oxidative stress, which can synergistically compromise its viability, underscoring the significance of pH stress in fungal pathogenicity [131]. Similarly, the pH-dependent toxicity of Congo Red on Aspergillus species illustrates how environmental pH influences fungal cell wall integrity and growth [138]. *Cryptococcus neoformans* adapts to the alkaline environment of the human host via the Pal/Rim pathway, with the Rra1 protein playing a crucial role in sensing and responding to pH changes [139]. Research on *Penicillium* sp. under acidic conditions has revealed the production of bioactive compounds, suggesting that pH stress also influences fungal metabolite production [135]. The pH signaling pathway is pivotal for maintaining cellular homeostasis across diverse fungi, impacting metabolism and pathogenicity [140]. In *Candida glabrata*, the transcription factor CgRds2 mediates the response to low pH stress, affecting energy metabolism and membrane permeability [141]. 

The Pal/Rim pathway, initially identified in *S. cerevisiae* and *A. nidulans*, has been extensively explored in fungal pathogens such as *Candida albicans, Cryptococcus neoformans*, and *Aspergillus fumigatus*. This alkaline pH-triggered signaling pathway regulates the expression of numerous genes essential for virulence. For example, in *C. albicans* transitioning from the host’s neutral pH to alkaline pH, the Rim signaling pathway activates genes involved in the yeast-to-hyphal transition, critical for tissue invasion. Similarly, the Rim pathway is crucial for tissue invasion and pathogenesis in *A. fumigatus* [142].

While extensively studied in fungi from the Ascomycete phylum, including *S. cerevisiae*, *C. albicans*, *A. nidulans*, and *A. fumigatus*, the Rim pathway’s conservation extends to more distantly related fungi such as the basidiomycetes *C. neoformans* and *U. maydis*. Despite the conservation of many Pal/Rim pathway signaling elements, including the endosomal sorting complex required for transport machinery, upstream signaling components such as the pH sensor are absent in the genomes of most basidiomycetes, as determined through direct sequence similarity [75,142].

## 8. Dimorphism

Dimorphism in fungi, characterized by their ability to exist in two distinct morphological forms, commonly as yeast and hyphal (or mycelial) phases, plays a pivotal role in their pathogenesis, environmental adaptation, and lifecycle, with significant implications for human health, agriculture, and the environment. The transition between these forms can be influenced by environmental factors such as temperature, with many pathogenic fungi demonstrating a thermally dimorphic switch essential for their virulence and survival in diverse hosts or conditions [143]. The interaction between dimorphism, pH, and pathogenicity in fungi is key to understanding their growth and virulence [77,144,145,146]. Fungal pathogens such as *Fusarium oxysporum* and *Penicillium marneffei* exhibit dimorphic growth, switching between yeast and hyphal forms in response to environmental signals such as temperature and pH [147]. pH plays a crucial role in regulating MAPK signaling, influencing fungal growth and pathogenicity. Furthermore, the ability of dimorphic fungi to alternate between morphologies is linked to their pathogenic potential, with thermally dimorphic fungi causing millions of infections each year. Understanding how pH affects dimorphic transitions and pathogenicity can provide key insights for managing fungal infections and developing targeted antifungal strategies.

Fungal dimorphism is also of evolutionary interest, serving as a crucial adaptation mechanism enabling fungi to colonize various ecological niches. The development of multicellular body structures in fungi, characterized by intermittent increases in phenotypic diversity, mirrors evolutionary processes observed in other kingdoms, suggesting a common mode of multicellular evolution across different life forms [148]. This evolutionary perspective is complemented by research on the diversity of fungal morphologies, arising from a combination of polar growth, cell division, and cell fusion, further highlighting the remarkable adaptability and complexity of fungal organisms [149].

Studies on different fungi have revealed that changes in surrounding pH levels can trigger a shift between these morphological states, thereby impacting their viability, growth, and ability to cause disease. For example, *Saccharomycopsis fibuligera* transitions from yeast to hyphal form under acidic conditions, accompanied by reduced production of carbohydrate-hydrolyzing enzymes and significant alterations in gene expression associated with enzyme production and pH adaptation [150]. These findings suggest that acidic environments can induce morphological alterations and influence metabolic processes in fungi.

Conversely, in *Ustilago maydis*, a gene encoding a Rho-like protein was identified as part of the Pal/Rim pathway, critical for responding to changes in environmental pH levels. Mutants lacking this gene exhibited diminished growth rates at alkaline pH and heightened sensitivity to ionic and osmotic stresses. However, no significant change in the transition from yeast to mycelial form induced by acidic pH was observed [62]. This suggests that while some fungi respond to acidic conditions with dimorphic transitions, others are more sensitive to alkaline environments.

Moreover, the significance of the transcription factor PacC, which regulates the pH signaling pathway in fungi, is evident in *Trichothecium roseum*. The expression of TrPacC increases with rising pH levels from 3 to 5, correlating with fungal growth, but decreases at higher pH levels, indicating its involvement in regulating growth and development in response to pH fluctuations [151]. This transcription factor is also implicated in activating genes associated with pathogenesis and resistance to stress in *Gaeumannomyces tritici*, exhibiting variations within the species in response to different pH conditions [152].

The study of GPCRs in the context of fungal dimorphism has revealed that these receptors can influence morphological changes through various signaling cascades. For instance, in *Aspergillus nidulans*, specific GPCRs have been shown to mediate glucose sensing, which in turn regulates cAMP signaling, promoting germination and hyphal growth while negatively affecting sexual development in a light-dependent manner [18]. Similarly, in *Aspergillus fumigatus*, GPCRs GprM and GprJ are implicated in the regulation of melanin production and the cell wall integrity pathway, affecting growth and virulence [153]. Moreover, the interaction between GPCRs and the RACK1 scaffolding protein in *Neurospora crassa* highlights the complexity of GPCR-mediated signaling in regulating fungal development. RACK1 interacts with G-protein signaling components, influencing hyphal growth, conidiation, and fertility, further underscoring the role of GPCRs in fungal dimorphism [154].

Fungal G-protein-coupled receptors (GPCRs) are critical in facilitating host–pathogen interactions by detecting environmental signals [155,156]. These receptors help fungi adjust their cellular functions, metabolism, and growth in response to various external cues, thus enhancing their survival, dissemination, and pathogenic properties [91]. Brown et al. [13] in Fusarium species that cause Fusarium Head Blight (FHB), GPCRs are vital for the pathogenic process, with receptors being indispensable for initiating infections in wheat [157]. Overall, fungal GPCRs are essential in host–pathogen exchanges, helping to modulate fungal responses to environmental triggers and influencing their pathogenic potential.

In fungi, signaling pathways such as protein kinase A (PKA) and mitogen-activated protein kinase (MAPK) play key roles in regulating dimorphism, a phenomenon that allows fungi to switch between different morphological forms. These pathways are tightly integrated and can interact with each other to coordinate the fungal response to changes in environmental conditions, including pH [157]. 

Despite the recognized relevance of the PKA and MAPK pathways in regulating fungal dimorphism in response to pH, a specific pH receptor that communicates with these pathways has not yet been identified in any model dimorphic fungus. The discovery of such a receptor could provide a more detailed understanding of how fungi sense and react to changes in environmental pH, opening new possibilities for more effective therapeutic interventions in the treatment of fungal diseases.

## 9. Perspectives

The identification of GPCRs responsible for pH-related signal transduction and dimorphism in fungi has been a challenge in scientific research. This is due to several reasons. Firstly, the diversity of fungi is immense, and different species may have unique and highly specialized signaling systems. This makes it difficult to generalize and find specific GPCRs for these events in all fungal species. Secondly, the function of many GPCRs in fungi has not been fully characterized, and their identification may require a multifaceted approach that combines genomics, proteomics, and functional analysis. Additionally, some GPCRs may have redundant functions or be involved in multiple signaling pathways, complicating their identification and specific study. Lastly, the regulation of pH and dimorphism in fungi is a highly dynamic and environment-dependent process. The GPCRs involved may be highly sensitive to changing environmental conditions, making their detection under standard laboratory conditions challenging.

Despite these challenges, ongoing research in this field is crucial, as understanding the GPCRs responsible for pH and dimorphism signaling in fungi could have significant applications in agriculture, biotechnology, and medicine, opening new possibilities for the control of fungal diseases and the manipulation of signaling pathways in these microorganisms. There are still some key areas that require further research and knowledge. Some of the aspects that still need to be studied include the following:(I)Identification of new GPCRs: Despite the progress made in identifying GPCRs in fungi, there are likely other receptors yet to be discovered. Searching for and characterizing new GPCRs would provide a more comprehensive view of the signaling pathways and biological functions regulated by these receptors;(II)Signaling pathway specificity: A deeper understanding is needed of how different GPCRs in fungi activate specific signaling pathways. This is crucial to understanding how these receptors coordinate diverse biological and adaptive responses to different stresses;(III)Interaction with other proteins: Studying the interactions between GPCRs and other intracellular proteins would provide a better understanding of how cellular responses are regulated and how these interactions may vary depending on the biological context;(IV)Specific biological functions: It is essential to determine the specific biological functions of GPCRs in different fungal species. Understanding how these receptors are involved in growth, reproduction, pathogenicity, and other physiological functions is essential to understanding their impact on fungal biology;(V)Therapeutic Potential: Research on GPCRs in fungi may have therapeutic implications, as some of these receptors may be targets for the development of new antifungal agents or treatments for fungal diseases.

## 10. Conclusions

G-protein-coupled receptors (GPCRs) have a significant impact on many areas due to their fundamental role in cellular signaling. In medicine, for example, GPCRs are important drug targets as they regulate a wide variety of biological processes from neurotransmission to hormone and neurotransmitter regulation. By modulating the activity of GPCRs, it is possible to influence these functions and treat various diseases, including neuropsychiatric disorders, cancer, cardiovascular diseases, and more. In phytopathology, research on GPCRs in fungal and bacterial pathogens has revealed their role in host–pathogen interactions. Understanding how GPCRs enable pathogens to adapt and manipulate host plant responses is crucial for the development of disease control strategies in agriculture and crop protection. In fungi, GPCRs play a critical role in adapting to various environmental stresses, enabling these organisms to survive and thrive in diverse conditions. The identification and characterization of new GPCRs and their interactions with other key proteins should be prioritized, as well as the study of specific GPCRs in pathogenic fungi and their potential as therapeutic targets. In conclusion, GPCRs are key players in cellular biology, and their study has important implications in both medicine and phytopathology to improve human health and agricultural production. Future research should focus on identifying new GPCRs, understanding the specificity of signaling pathways activated by different GPCRs, studying the interactions between GPCRs and other intracellular proteins, and exploring the therapeutic potential of targeting GPCRs in fungi. Additionally, understanding how GPCRs in fungi adapt and respond to different types of stress is crucial to understanding how these microorganisms adapt to different environments.

## Figures and Tables

**Figure 1 jof-11-00213-f001:**
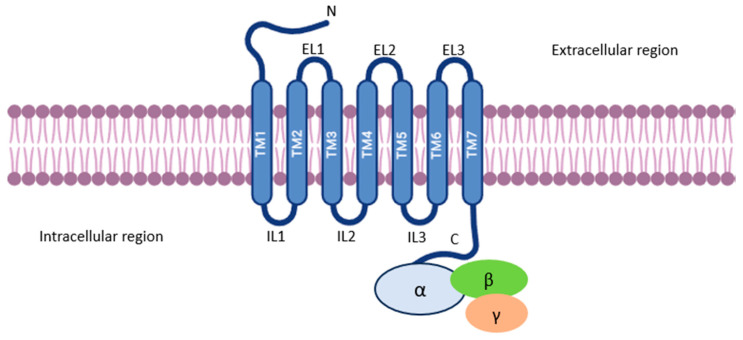
Structural blueprint of GPCR. GPCR structure features seven transmembrane helices (TM1–TM7) spanning the outer membrane. Three extracellular loops (ECL1–ECL3) contain conserved cysteine residues forming disulfide bonds for stability. Three intracellular loops (ICL1–ICL3) interact with G-proteins for signal transduction. The N-terminus is extracellular, and the C-terminus is intracellular, showing the receptor’s membrane orientation.

**Figure 2 jof-11-00213-f002:**
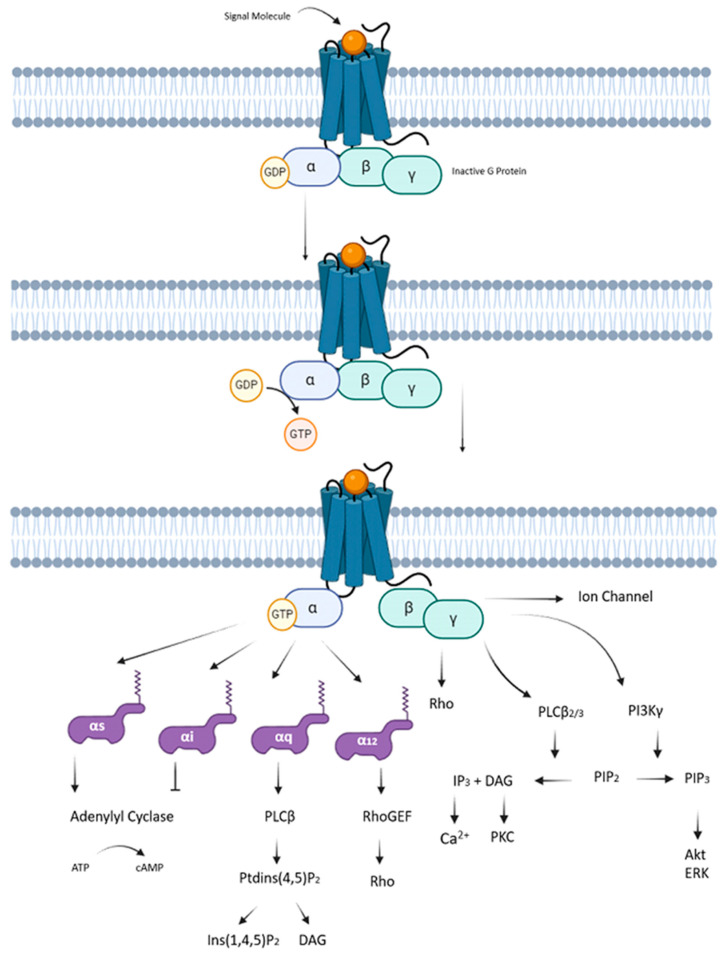
Interaction and differential functions of Gα and Gβγ subunits in the signaling of GPCRs. When a GPCR is activated by a ligand, the resting state of the G-protein bound to GDP is disrupted. Upon activation, the Gα subunit, now bound to GTP, dissociates from the Gβγ dimer. Various Gα subunits (Gαs, Gαi, Gαq, Gα12/13) and their interactions with key targets such as adenylate cyclase, phospholipase C, and Rho family GTPases are shown. Additionally, the interactions of the Gβγ subunit with kinases, regulation of ion channels, and its influence on other GPCRs and small GTPases are highlighted.

**Figure 3 jof-11-00213-f003:**
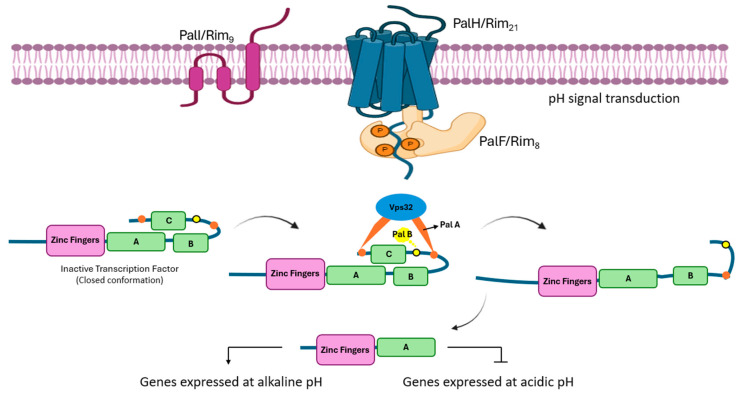
Alkaline pH activation of the Pal/Rim pathway in Ascomycota. PalH/Rim21, a receptor with seven transmembrane (7TM) domains, senses pH changes and signals through its C-terminal tail to PalF/Rim8, an arrestin. PalA/Rim20 binds to YPXL motifs on the PacC transcription factor, which is cleaved by PalB/Rim13. This cleavage allows PacC to be further processed, enabling it to activate genes at alkaline pH and repress them at acidic pH. A, B, and C represent C-terminal residues that undergoes a two successive proteolytic cleavages to activate PacC.

**Figure 4 jof-11-00213-f004:**
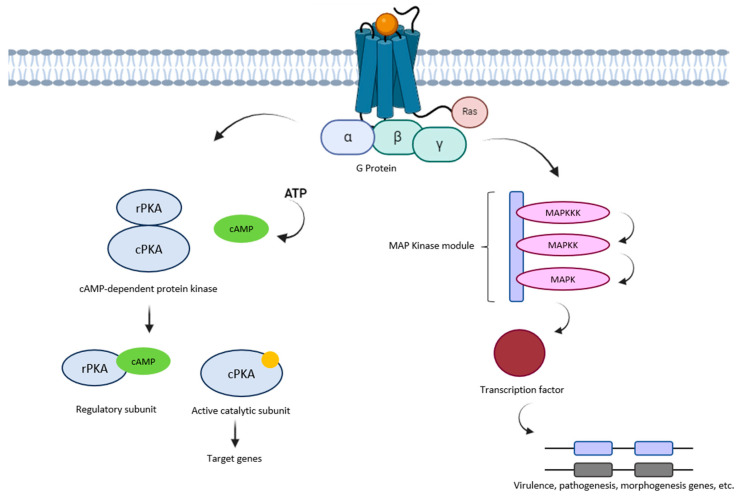
MAPK and PKA signaling pathways. The MAPK pathway responds to external stimuli such as stress (osmotic, oxidative) through a cascade involving MAPKKKs, MAPKKs, and core MAPKs (Pmk1, Hog1, Slt2). This pathway regulates hyphal development, stress responses, virulence, and secondary metabolite production. In contrast, the PKA pathway is activated by extracellular signals detected by GPCRs, leading to cAMP production by adenylate cyclase (AC). cAMP activates PKA, which phosphorylates downstream targets, including transcription factors, to regulate metabolic processes, growth, and stress responses within fungal cells.

**Table 2 jof-11-00213-t002:** Classification of GPCR in fungi.

Class	Description	Characteristics	Examples	Ref.
I and II	Pheromone Receptors	-Similar to Ste2 and Ste3 receptors, sensing α-factor and a-factor, respectively. -Expressed on a-cells and α-cells.	GprA, GprB (*Aspergillus fumigatus*)	[36,37]
III	Carbon Receptors	Involves the third intracellular loop and cytoplasmic tail in G-protein binding and receptor desensitization.	GprC, GprD *(A. fumigatus)*	[36,38]
IV	Putative Nitrogen Receptors	-Characterized by a PQ loop containing two repeats spanning two TM helices. -Possibly involved in transporting cysteine and cationic amino acids.	GprF. GprG, GprJ *(A. fumigatus)*	[39,40,41]
V	Putative Carbon and Amino Acid Receptors (cAMP receptor-like)	A unique organization with a Git3-carbon-sensing domain from S. pombe, CrlA-cAMP receptor domain from Dictyostelium discoideum, and extended cytoplasmic tail.	GprV, Gpr H (*Aspergillus nidulans)*	[12,13,18,42]
VI	RGS-Domain-Containing Receptors	Contain an intracellular RGS domain.		[43]
VII	Orthologues of MG00532 with Weak Similarity to Rat Growth-Hormone-Releasing Factor	Show limited similarity to the rat growth hormone-releasing hormone receptor.	GprM (*Aspergillus nidulans)*, Gpr8 (*Trichoderma reesei)*	[18,32,44,45]
VIII	Mammalian Progesterone Receptor-Like Receptors	Resemble mPRs that activate inhibitory G-proteins, suggesting their role as GPCRs	MoRgs7 (*Magnaphorthe oryzae*)	[32,44]
IX	Microbial Opsin Receptors	Functionally conserved, with FfCarO opsin receptor as a green-light-driven proton pump influencing hyphal development.	NopA (*Aspergillus fumigatus)*	[37,46]
X	Family Pth11 Receptors	The CFEM domain of the M. oryzae class X Pth11 receptor is critical for hydrophobic sensing	BBA_00828, BBA_05185 (*B. bassiana)*, CmEST-463 (*C. minitans)*	[47,48,49,50]

**Table 3 jof-11-00213-t003:** Modified classifications of GPCR in fungi.

Class	Description	Characteristics	Examples	Ref.
VI	GrpK-like/RGS Domain Receptors	Contain RGS domains, suggesting potential regulatory functions.	GprK (*Aspergillus fumigatus)*	[37]
VII	Rat Growth Hormone-Releasing Factor Receptor-Like Receptors	Exhibit similarities to rat growth hormone-releasing factor receptors.	GprM (*Aspergillus fumigatus)*	[37]
VIII	mPR-like/PAQR Receptors	Share characteristics with mammalian progesterone receptors.	GprO, GprP (*Aspergillus fumigatus)*	[37]
X	Lung 7TM Superfamily Receptors	Represent receptors similar to the Lung 7TM Superfamily.		[37]
XI	GPCR89/ABA GPCR Receptors	Belong to the GPCR89/ABA GPCR class. Could be involved in signaling related to abscisic acid, a plant hormone.	Gpr-12 (*N. crassa)*	[52]
XII	Family C-like Receptors	Similar to receptors found in Family C, which include metabotropic glutamate receptors.	Gpr-13 (*N. crassa)*	[52]
XIII	DUF300 Superfamily/PsGPR11 Receptors	Associated with the DUF300 Superfamily and PsGPR11.	Gpr-14 (*N. crassa)*	[52]
XIV	Pth11-Like Receptors	Related to the Pth11 family of receptors, involved in signaling and pathogenicity in plants.	Gpr-15, Gpr-23, Gpr-29 (*N. crassa)*	[52]

**Table 4 jof-11-00213-t004:** MAPK main functions in fungi.

Organism	MAPKs	Main Functions	Ref.
*Aspergillus fumigatus*	MpkA, MpkC, SakA, MpkB	Mpka: Regulates cell wall integrity. MpkC and SakA: Response to various types of stress (osmotic, oxidative, adaptation to high temperatures), cell wall damage, and virulence. MpkB: function yet uncharacterized.	[80]
*Cytospora chrysosperma*	CcPmk1, CcHog1, CcSlt2	Core regulators of fungal pathogenicity, development, and stress responses; required for responses to hyperosmotic pressure, cell wall inhibition agents, and oxidative stress; CcPmk1 and CcSlt2 essential for hyphal growth and fungal pathogenicity.	[17]
*Saccharomyces cerevisiae*	MAPKs: Fus3; DUSPs: Sdp1, Msg5	Crucial for modulating MAPK signaling flow; affects cell physiology and pathogenicity in various fungi.	[74]
*Candida albicans*	FUS3 (homolog)	Loss of mating efficiency and decreased biofilm formation.	[80]
*Cryptococcus neoformans*			
*Aspergillus flavus*	Fus3	Regulates aflatoxin biosynthesis by modulating substrate levels; demonstrates the role of MAPKs in secondary metabolism.	[74]
*Aspergillus nidulans*	Mpk8	Inhibition of sexual reproduction and impact on the production of secondary metabolites such as sterigmatocystin, terrequinone A, and penicillin.	[80]
*Fusarium oxysporum*	Fmk1	Utilized in response to host alkalinization, highlighting the role of MAPKs in fungal pathogenicity through environmental sensing.	[81]
*Clonostachys chloroleuca*	Crmapk	Significant for mycoparasitism and biocontrol activities; investigated through yeast two-hybrid screening to identify interacting proteins.	[82]
*Aspergillus nidulans*	Pheromone module	Regulates development and secondary metabolism; HamE proposed as a scaffold protein for this pathway	[83]
Magnaporthe oryzae	Pmk1, Mps1	Essential for appressorium formation, penetration, and invasive growth; critical for fungal infection processes	[77]

## Data Availability

No new data were created or analyzed in this study. Data sharing is not applicable to this article.

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
