# Peer review of "Fungal Stress Responses and the Importance of GPCRs"

_jof, 2025, doi:10.3390/jof11030213_

Round 1
Reviewer 1 Report
In the review article entitled Fungal Stress and the Importance of GPCPCR the authors give the overview of signalling pathways working via GPCR, G-proteins and different downstream signalling pathways, which play important roles in fungal stress responses. Although the topic of the article is interesting and the title of the article promising, it is not focused enough and the chapters are poorly structured. Introduction as well as Conclusions need much more in depth elaboration. Description of GPCR and G-proteins and signalling pathways that include GPCR are at some point too detailed. There are some confusion in sub-chapters and some necessary repetitions.
In chapter 6.1.1. the authors state that HOG signalling pathway plays a crucial role in stress conditions related to salt concentration, but in Table 5.:Fungi with Mechanisms coping with osmotic stress, they completely left out halotolerant fungi that have been extensively studied in this respect.
Author Response
Comments 1: In the review article entitled Fungal Stress and the Importance of GPCPCR the authors give the overview of signalling pathways working via GPCR, G-proteins and different downstream signalling pathways, which play important roles in fungal stress responses. Although the topic of the article is interesting and the title of the article promising, it is not focused enough and the chapters are poorly structured. Introduction as well as Conclusions need much more in depth elaboration. Description of GPCR and G-proteins and signalling pathways that include GPCR are at some point too detailed. There are some confusion in sub-chapters and some necessary repetitions.
Response 1: Thank you for your feedback and for taking the time to review our manuscript. We acknowledge your concerns regarding the focus and structure of the chapters, as well as the need for more in-depth elaboration in the introduction and conclusions. We refine the focus of each chapter and improve the overall structure to ensure a more cohesive and logical flow of information. Also, we expanded the introduction to provide a more comprehensive background, clearly outlining the objectives and significance of the review. The conclusions were restructured to provide a more thorough summary of the key findings and their implications, highlighting future research directions. We have noted your concerns regarding the descriptions of GPCRs, G-proteins, and signaling pathways being overly detailed, as well as the confusion and repetitions in the sub-chapters. We have simplified and condensed the descriptions of GPCRs, G-proteins, and their signaling pathways to focus on the most relevant details. The chapter numbering was incorrect, with two subtopics labeled as number 2. This has been corrected, and we have reorganized the sub-chapters to improve clarity and ensure a logical flow of information. Unnecessary repetitions have been identified and removed to enhance readability and coherence. We believe these changes address your concerns and improve the overall quality of the manuscript.
Comments 2: In chapter 6.1.1 the authors state that HOG signalling pathway plays a crucial role in stress conditions related to salt concentration, but in the Table 5. Fungi with mechanisms coping with osmotic stress, they completely left out halotolerant fungi that have been extensively studied in this respect.
Response 2: Thank you for pointing this out. In response to your observation, we have made the following revisions to address the omission of halotolerant fungi: We have included the halotolerant fungi Hortaea werneckii, Wallemia ichthyophaga, and Debaryomyces hansenii, which are well-studied for their mechanisms in coping with high salt concentrations. These fungi have been integrated into Table 5 to provide a more comprehensive view of the HOG signaling pathway and its role under osmotic stress conditions. Additionally, while Aspergillus species are not specifically known for their halotolerance, they can survive and produce mycotoxins under osmotic stress. Therefore, we have also included examples of Aspergillus in the table to illustrate their resilience and response to osmotic stress. Regarding Scedosporium, although it is not halotolerant, we have decided to include relevant information about its stress responses to provide a broader context. We hope these additions and revisions adequately address your concerns and enhance the comprehensiveness of our manuscript.
Reviewer 2 Report
This is a comprehensive review on a timely topic. G protein coupled receptors (GPCRs) are needed for a huge array of responses to environmental and other signals. In medicine they are among the most important drug targets, but progress in targeting the signaling pathways of fungal pathogens still awaits better understanding of what the ligands and effectors are.
1. line 202-203 - The sentence is incomplete, but more importantly, if I understand correctly, addresses both Tables 2 and 3. Could revise; "As we can see in Tables 2 and 3 .. the Classical ... include classes I-V [or as appropriate] ... ... and the Novel ... include classes ...
Classes V and VI look the same in Table 3 and in Table 2, then the classifications diverge. The text cites reference 52 as the source for the modification but this is not completely clear; I would suggest to revise this part, clarifying a little better what was modified in the going from Table 2 to Table 3, why, and by whom (all in reference 52? by the authors of this review?).
2. lines 263-264 - PalH is not strictly a GPCR as its downstream partner is arrestin rather than a heterotrimeric G protein. In my opinion it is fine to include, and indeed is a really important one, because the PacC pathway is unique to fungi. It would be good, though to explain that it is a 7TM sensor but not G protein coupled. Lines 302-305 would also need to be reworded to reflect this. The title of Figure 3 belongs to Figure 2; here again would be a good place to emphasize pH-sensing by a 7TM membrane sensor coupled to arrestin.
3. There should still be time to do some updating before publication. For example, two GPCR papers just out, on nematode trapping fungi: https://pubmed.ncbi.nlm.nih.gov/38877225/ https://pubmed.ncbi.nlm.nih.gov/38649409/
4. line 422 - section 6 - adaptation - in signaling pathways, adaptation also means that within a certain range, output depends on the change in signal rather than the level of the signal (as in bacterial chemotaxis). Here, adaptation means tolerance to stress or a reprogramming of gene expression in response to a stress or signal. It would be helpful to define this just once at the start of section 6.
5. just a suggestion (might expand the range of the review too much) - to briefly discuss fungal opsins, and the rhodopsin-guanylyl cyclases of zoospore-forming fungi. Though not considered true fungi by many, the recent progress is exciting enough to be a welcome addition - again, if this doesn't go beyond the scope.
6. along the lines of comment 2: CFEM domain containing GPCRs - is there evidence that the signals from these are transduced by heterotrimeric G proteins? These being fungal-specific, it seems important to discuss.
Author Response
Comments 1: line 202-203 - The sentence is incomplete, but more importantly, if I understand correctly, addresses both Tables 2 and 3. Could revise; "As we can see in Tables 2 and 3 .. the Classical ... include classes I-V [or as appropriate] ... ... and the Novel ... include classes ...Classes V and VI look the same in Table 3 and in Table 2, then the classifications diverge. The text cites reference 52 as the source for the modification but this is not completely clear; I would suggest to revise this part, clarifying a little better what was modified in the going from Table 2 to Table 3, why, and by whom (all in reference 52? by the authors of this review?).
Response 1: Thank you for highlighting the issue with lines 202-203 and the need for clarification regarding Tables 2 and 3. In lines 182-199, we explain Table 2, detailing the existing classification of GPCRs. Martin et al., in their review, identified several sources that suggested new groups, which we included in Table 3. Therefore, we cited Martin et al., and the tables include the references he used. However, we recognize a potential error in lines 200 to 203, where we refer to Table 3 but describe categories 1 to 5, while Table 3 starts from category 6. This discrepancy could be confusing. We have modified the order of the tables to improve the flow of the text. The potential error previously noted in lines 200 to 203 has been corrected and can now be seen in lines 217 to 220 of the revised manuscript.
Comments 2: lines 263-264 - PalH is not strictly a GPCR as its downstream partner is arrestin rather than a heterotrimeric G protein. In my opinion it is fine to include, and indeed is a really important one, because the PacC pathway is unique to fungi. It would be good, though to explain that it is a 7TM sensor but not G protein coupled. Lines 302-305 would also need to be reworded to reflect this. The title of Figure 3 belongs to Figure 2; here again would be a good place to emphasize pH-sensing by a 7TM membrane sensor coupled to arrestin.
Response 2: Thank you for pointing this out. We agree with this comment, therefore, we have made some modification to the paragraph, addressing the comments for this point (lines 278 to 292). Also, text modification was made to the figure title because, indeed, the titles were repeated. Similarly, the description of the figure was updated to specify details about arrestin:
Comments 3: There should still be time to do some updating before publication. For example, two GPCR papers just out, on nematode trapping fungi: https://pubmed.ncbi.nlm.nih.gov/38877225/ https://pubmed.ncbi.nlm.nih.gov/38649409/
Response 3: Thank you for your suggestion. We appreciate your recommendation to include recent papers on GPCRs in nematode-trapping fungi. We integrate the findings from the suggested publications on line 236 to 246. Thank you for helping us enhance the manuscript.
Comments 4: line 422 - section 6 - adaptation - in signaling pathways, adaptation also means that within a certain range, output depends on the change in signal rather than the level of the signal (as in bacterial chemotaxis). Here, adaptation means tolerance to stress or a reprogramming of gene expression in response to a stress or signal. It would be helpful to define this just once at the start of section 6.
Response 4: Thank you for pointing this out. We agree with this comment, therefore, we have made some modification to the paragraph, addressing the comments for this point (lines 447 to 454)
Comments 5: just a suggestion (might expand the range of the review too much) - to briefly discuss fungal opsins, and the rhodopsin-guanylyl cyclases of zoospore-forming fungi. Though not considered true fungi by many, the recent progress is exciting enough to be a welcome addition - again, if this doesn't go beyond the scope.
Response 5: Thank you for your suggestion regarding the inclusion of fungal opsins and rhodopsin-guanylyl cyclases of zoospore-forming fungi. While we appreciate the relevance and excitement surrounding these topics, incorporating them would significantly expand the scope of our review. Our focus remains on GPCRs in fungi, and including such specialized topics might dilute the core content.
Comments 6: along the lines of comment 2: CFEM domain containing GPCRs - is there evidence that the signals from these are transduced by heterotrimeric G proteins? These being fungal-specific, it seems important to discuss.
Response 6: Thank you for pointing this out. We agree with this comment, therefore, we have made some modification to the paragraph, addressing the comments for this point (lines 290 to 292)
Round 2
Reviewer 1 Report
The authors give overview of signalling pathways working via GPCR, G-proteins and different downstream signalling pathways together with the overview of fungal responses to different environmental stresses. As GPCR play crucial role in fungal biology this review is of interest of scientists working on fungal biolgy, particularly on fungal responses to different stresses and represents important contribution.
In the revised version of the manuscript the authors answered my comments: text is improved, more focused and in line with the title of this review article and in Table 5 they included halotolerant fungi with appropriate citations.
The numbers of subtitles in chapter 7 should be corrected: 7.1.2 is followed by 7.1.4 instead of 7.1.3